# Clinical predictors of encephalitis in UK adults–A multi-centre prospective observational cohort study

**Sylviane Defres**[1,2,3,4]*, **Kukatharmini Tharmaratnam**[5], **Benedict D. Michael**[1,2,6], **Mark Ellul**[1,3,6], **Nicholas W. S. Davies**[7], **Ava Easton**[1,8], **Michael J. Griffiths**[1,9], **Maneesh Bhojak**[6], **Kumar Das**[6], **Hayley Hardwick**[1], **Chris Cheyne**[5], **Rachel Kneen**[1,9], **Antonieta Medina-Lara**[10], **Anne Christine Salter**[11], **Nicholas J. Beeching**[2,3,4], **Enitan Carrol**[1,12], **Angela Vincent**[13], **ENCEPH UK study group**[¶], **Marta Garcia-Finana**[5ↄ], **Tom Solomon**[1,3,6ↄ]

**1** Department of Clinical Infection Microbiology and Immunology, Institute of Infection, Veterinary and Ecological Sciences, University of Liverpool, Liverpool, United Kingdom, **2** Tropical and Infectious Disease Unit, Liverpool University Hospitals NHS Foundation Trust, Liverpool, United Kingdom, **3** National Institute for Health Research (NIHR) Health Protection Research Unit (HPRU) in Emerging and Zoonotic Infections, Institute of Infection, Veterinary and Ecological Sciences, University of Liverpool, Liverpool, United Kingdom, **4** Department of Clinical Sciences, Liverpool School of Tropical Medicine, Liverpool, United Kingdom, **5** Department of Biostatistics, Institute of Translational Medicine, University of Liverpool, Liverpool, United Kingdom, **6** Department of Neurology, The Walton Centre NHS Foundation Trust, Liverpool, United Kingdom, **7** Department of Neurology, Chelsea and Westminster NHS Trust, London, United Kingdom, **8** Encephalitis Society, Malton, United Kingdom, **9** Department of Paediatric Neurology, Alder Hey Hospital Children's NHS Foundation Trust, Liverpool, United Kingdom, **10** Health Economics Group, University of Exeter Medical School, Exeter, United Kingdom, **11** Patient Representative, Encephalitis Society, Malton, United Kingdom, **12** Department of Paediatric Infectious Diseases, Alder Hey Hospital Children's NHS Foundation Trust, Liverpool, United Kingdom, **13** Nuffield Department of Clinical Neurosciences, University of Oxford, Oxford, United Kingdom

ↄ These authors contributed equally to this work.
¶ Membership of the ENCEPH UK study group is provided in the Acknowledgments.
* sdefres@liverpool.ac.uk

**Data Availability Statement:** All relevant data are within the paper and its Supporting Information files.

## Abstract

### Objectives

Encephalitis, brain inflammation and swelling, most often caused by an infection or the body's immune defences, can have devastating consequences, especially if diagnosed late. We looked for clinical predictors of different types of encephalitis to help clinicians consider earlier treatment.

### Methods

We conducted a multicentre prospective observational cohort study (ENCEPH-UK) of adults (> 16 years) with suspected encephalitis at 31 UK hospitals. We evaluated clinical features and investigated for infectious and autoimmune causes.

### Results

341 patients were enrolled between December 2012 and December 2015 and followed up for 12 months. 233 had encephalitis, of whom 65 (28%) had HSV, 38 (16%) had confirmed

**Funding:** This study was funded by the National Institute for Health Research (NIHR) under its Programme Grants for Applied Research scheme (grant number RP-PG-0108-10048) awarded to TS. The funders of the study had no role in study design, data collection, analysis or interpretation or writing of the report.

**Competing interests:** The authors have declared that no competing interests exist.

or probable autoimmune encephalitis, and 87 (37%) had no cause found. The median time from admission to 1st dose of aciclovir for those with HSV was 14 hours (IQR 5–50); time to 1st dose of immunosuppressant for the autoimmune group was 125 hours (IQR 45–250). Compared to non-HSV encephalitis, patients with HSV more often had fever, lower serum sodium and lacked a rash. Those with probable or confirmed autoimmune encephalitis were more likely to be female, have abnormal movements, normal serum sodium levels and a cerebrospinal fluid white cell count < 20 cells x$10^6$/L, but they were less likely to have a febrile illness.

## Conclusions

Initiation of treatment for autoimmune encephalitis is delayed considerably compared with HSV encephalitis. Clinical features can help identify patients with autoimmune disease and could be used to initiate earlier presumptive therapy.

## Introduction

Encephalitis is inflammation and swelling of the brain most often caused by an infection, or by the body's immune defences [1] Patients typically present with altered consciousness, which may range from mild behavioural abnormality to deep coma. Although there is a long and growing list of potential microbial aetiologies, herpes simplex virus (HSV) remains the most common sporadic infectious cause [2–5]. During the last fifteen years, non-infectious immune-mediated causes of encephalitis have also been recognised with increasing frequency, especially associated with antibodies against the N-methyl-D-aspartate receptor (NMDAR), leucine-rich glioma inactivated 1 (LGI-1) protein which is part of the voltage gated potassium channel complex and glutamic acid decarboxylase (GAD) [6–9].

The outcome of encephalitis is improved with prompt recognition and initiation of appropriate treatment. Several studies have shown that for HSV encephalitis, starting the antiviral drug aciclovir within 48 hours of hospital admission is associated with a better outcome [10]. The drug is therefore typically started as soon as encephalitis is suspected. In NMDAR encephalitis retrospective studies have shown that better outcomes are also associated with earlier immunomodulatory treatment [11]. However, identifying patients with encephalitis, and with particular aetiologies, can be difficult because of the non-specific nature of the symptoms, and overlap of the clinical features for the different causes [3–5, 12–14]. This leads to delays to delays in accessing the appropriate health care and receiving prompt investigation and management [15, 16]. We therefore studied the clinical features of adults presenting with suspected encephalitis and assessed which of the features were associated the different types of encephalitis, in particular HSV and the autoimmune forms, in order to prompt earlier treatment.

## Methods

### Study population and design

Patients were recruited from 31 hospitals across England, Wales and Scotland in a multicentre, prospective observational cohort study, which was part of a programme of studies by the National Institute for Health Research (NIHR) applied research programme grant on Understanding and improving the Outcome of encephalitis in the UK (ENCEPH-UK). Details of the whole programme can be found at www.encephuk.org.

Patients were eligible if they were aged 16 years or older with clinically suspected encephalitis defined as an acute or sub-acute alteration in consciousness, cognition, personality or behaviour changes for more than 24 hours, along with any two of fever, prodromal illness, new onset seizures, focal neurological signs, pleocytosis (cerebrospinal fluid [CSF] white cell count of $> 4$ cells x10$^6$/L), neuroimaging or electroencephalogram (EEG) compatible with encephalitis; [5] or any clinical suspicion of encephalitis when these investigations had not been completed at the time of recruitment and with no alternative diagnosis evident (Table 1).

Patients were followed up for 12 months, and the Glasgow Outcome Score (GOS) which ranges from 1 (death) to 5 (full recovery) recorded. The study protocols were approved by participating sites and the National Research Ethics Service (now part of the Health Research Authority) East Midlands Nottingham 1 committee (reference 11/EM/0442). Written consent for entry into the study was obtained from patients or an accompanying relative. Standardised case record forms for clinical, laboratory and radiological data were recorded on a secure online database (Openclinica$^{TM}$).

## Aetiological testing

All CSF samples had microscopy, culture and polymerase chain reaction (PCR) performed according to national guidelines in the admitting hospitals, regional diagnostic centres or at the University of Liverpool. This included standard first line testing of HSV-1 and 2, varicella zoster virus (VZV), enteroviruses and parechoviruses, and second line testing depending on the clinical presentations [17, 18]. Serum, and where available CSF, were tested for a panel of autoantibodies either at the hospital site, the University of Liverpool, or the University of Oxford [6, 7, 19]. All patients with suspected autoimmune encephalitis had, as a minimum, tests to detect NMDAR, VGKC-complex, glutamic acid decarboxylase (GAD) and

**Table 1. Inclusion and exclusion criteria for suspected encephalitis.**

| Suspected encephalitis | **Inclusion Criteria** |
|---|---|
| | *(A)* Acute or sub-acute (<4 weeks) alteration in consciousness, cognition, personality or behaviour* persisting for more than 24 hours |
| | • *Plus ANY two of*: |
| | • Fever ($\geq$ 38˚C) / Prodromal illness–acute or sub-acute |
| | • Seizures: New onset |
| | • Focal Neurological Signs–Acute or Sub-acute onset. |
| | • Including |
| | • Focal weakness |
| | • Oromotor dysfunction |
| | • Movement disorders** including Parkinsonism*** |
| | • Amnesia |
| | • Pleocytosis: Cerebrospinal fluid white cell count >4 cells/ul |
| | • Neuroimaging: Compatible with encephalitis |
| | • Electroencephalogram (EEG): compatible with encephalitis |
| | * personality / behaviour change including agitation, psychosis, somnolence, insomnia, catatonia, mood lability, altered sleep pattern and (in children): new onset enuresis, or irritability, |
| | ** Movement disorders includes chorea, athetosis, dystonia, hemiballismus, stereotypies, orolingual dyskinesia and tics |
| | ***bradykinesia, tremor, rigidity and postural instability |
| | OR |
| | *(B) Clinical suspicion of encephalitis but above investigations have not yet been completed* |
| | OR |
| | *(C) Clinical suspicion of encephalitis and the patient died before investigations completed* |
| | **Exclusion criteria** |
| | • Patients with non-infectious or non-immune central nervous system disorders due to hypoxic, ischaemic, vascular, toxic or metabolic causes |
| | • Patients with *pre-existing* indwelling ventricular devices |

paraneoplastic (onconeuronal) antibodies. The standard onconeuronal antibody panel included Hu(D), Yo, CV2/CRMP5, Ri, Ma1, Ma2, amphiphysin, Tr, SOX1, Zic4, titin, recoverin, PKCγ & Purkinje cell antibodies, tested by immunohistochemistry and recombinant immunoblot. Newer antibody tests that became available during the course of the study were incorporated prospectively; these included α-amino-3-hydroxy-5-methyl-4-isoxazolepropionic acid receptor (AMPAR), contactin associated protein–like 2 (CASPR2); dopamine 2 receptor (DR2), dipeptidyl-peptidase–like protein 6 (DPPX), γ-aminobutyric acid (GABA) type A and type B receptors, CV2, metabotropic glutamate receptor 5 (mGluR5), and LGI-1/contactin/CASPR2 testing to replace earlier VGKC-complex. Remaining samples from earlier patients were tested retrospectively if available (this was performed in 35 patients; 2 were classified as autoimmune encephalitis and 33 remained as encephalitis of unknown aetiology). Further analyses were performed in the unknown aetiological group with a history of foreign travel for extended infection screening at the Rare and Imported Pathogen Laboratory, Public Health England, Porton Down.

## Methods for analysis

Patients were classified into causes of encephalitis with different degrees of certainty using published case definitions [5–7, 20, 21]. They were reclassified when updated autoimmune case definitions became available (Table 2) [22]. Where the classification was unclear, cases were discussed by a panel of experts and a consensus reached.

There were two primary analyses; one looked for an association between baseline variables and HSV encephalitis, the other between baseline variables and autoimmune encephalitis. Univariate analysis was performed initially on each variable to look for associations using unpaired t-tests, Mann- Whitney U test, chi-square test and Fisher's exact test where appropriate. Clinical variables that had potential for significance were selected using the lasso variable selection approach to perform a multiple logistic regression model. The stepBIC approach to multiple logistic regression model was used to identify variables significantly associated with encephalitis. Data were analysed using R [23].

## Results

Between 2012 and 2015, 341 patients with suspected encephalitis were enrolled, of whom 233 (68%) met the case definition of clinically diagnosed encephalitis (Fig 1); 108 (32%) did not have encephalitis, but a range of mimicking conditions, most commonly septic encephalopathy 18 (17%). Of the 233 patients with encephalitis, 108 (46%) had an infectious cause including 65 (28%) with HSV type 1 or 2, 32 (14%) with other viruses and 11 (5%) with bacterial or fungi. Autoimmune encephalitis was diagnosed in 38 (16%) patients, of whom 35 (15%) had a confirmed diagnosis with autoantibodies and 3 (1%) had probable; in addition, 23 (10%) had possible autoimmune encephalitis making 61 in total. Sixty-four (27%) patients with encephalitis had no cause identified, despite further testing for infectious and autoimmune causes. Clinical features of the study population are shown in Table 3.

The median age of the 233 patients with encephalitis was 54 years (IQR 34–68) and 115 (59%) were female. 70 (30%) had at least one comorbidity. 27 (12%) patients had immunocompromise, which was significantly more likely in those with viruses other than HSV detected (9 [28%] of 32, versus 18 [9%] of 201, p<0.0001).

### HSV encephalitis

Patients with HSV encephalitis typically presented with a brief febrile illness (2 days [IQR 1–2]) and altered cognitive function, particularly confusion/disorientation or altered

**Table 2. Case definitions for types of encephalitis.**

| | |
|---|---|
| Clinically diagnosed Encephalitis | Patients with Suspected Encephalitis (defined above) with evidence of brain inflammation from surrogate markers (e.g. on brain imaging or CSF pleocytosis) and no alternative diagnosis made |
| Viral encephalitis | • Confirmed: Clinically diagnosed encephalitis AND positive CSF PCR for a viral pathogen or intrathecal antibody<br>• Probable: Encephalitis AND detection of an appropriate pathogen by either throat swab, rectal swab or serology |
| Bacterial or fungal encephalitis | Clinically diagnosed encephalitis AND detection of an appropriate bacterial or fungal pathogen from either blood or CSF by PCR, culture or gram stain |
| Progressive multifocal leukoencephalopathy (PML) | • Confirmed:<br>1) clinical features of PML including cognitive decline, speech and language deficits and gait abnormalities; AND<br>2) neuroimaging findings in keeping; AND<br>3) presence of JC virus in CSF<br>• Probable:<br>Two of the above criteria met |
| Autoimmune encephalitis | • Confirmed:<br>rapid progression short term memory loss, altered mental state or psychiatric symptoms AND detection of an appropriate autoimmune antibody from either blood or CSF<br>• Probable:<br>1) rapid progression short term memory loss, altered mental state or psychiatric symptoms;<br>2) exclusion of well-defined syndromes of autoimmune encephalitis (Acute disseminated encephalomyelitis, Bickerstaff etc); AND<br>3) absence of well characterised autoantibody in serum or CSF and at least 2 of: MRI abnormal in keeping with autoimmune encephalitis; CSF pleocytosis; brain biopsy inflammatory infiltrates and excluding other disorders; AND<br>4) reasonable exclusion of alternative causes<br>• Possible:<br>1) rapid progression short term memory loss, altered mental state or psychiatric symptoms; AND<br>2) at least one of the following:<br> a. New focal CNS findings<br> b. Seizures not explained by previously known seizure disorder<br> c. CSF pleocytosis (white cell count of more than 5 cells per mm3)<br> d. MRI features suggestive if encephalitis<br>3) absence of well characterised autoantibody in serum or CSF |
| Paraneoplastic encephalitis | Clinically diagnosed encephalitis; AND<br>Cancer diagnosed within 5 years of neurological symptoms development; AND<br>No antineuronal antibody detected. |
| Hashimoto's encephalitis | All 6 of the following criteria:<br>• encephalopathy with seizures, myoclonus, hallucinations or stroke like episodes;<br>• subclinical or mild overt thyroid disease (usually hypothyroidism);<br>• brain MRI normal or with non-specific abnormalities;<br>• presence of serum thyroid antibodies (thyroid peroxidase, thyroglobulin);<br>• absence of well characterized neuronal antibodies in serum and CSF;<br>• reasonable exclusion of alternative cause |

personality/behaviour. Compared with the 168 other encephalitis patients, the 65 with HSV encephalitis were significantly older (median 60 years [IQR 47–71] versus 52 [32–66] years, p = 0.004) and more likely to have a fever on examination (47 [75%] of 65 versus 55 [33%] of 168, p = 0.001), but less likely to have a history of agitation (13 [20%] versus 76 [45%], p = 0.001), or a rash (4 [6%] versus 29 [17%], p = 0.04). Investigations showed the HSV

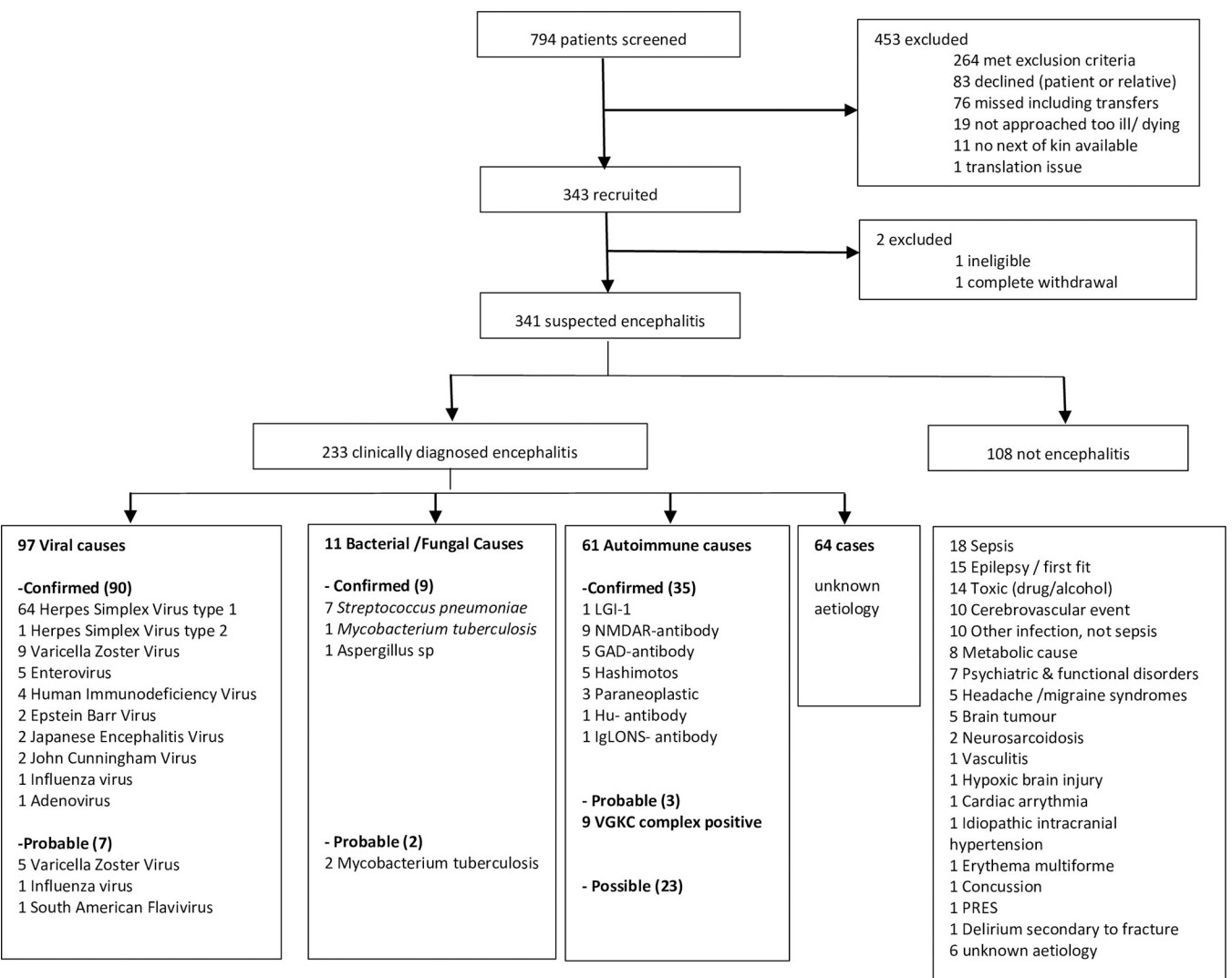

PRES is posterior reversible encephalopathy syndrome

**Fig 1. Flow diagram of recruitment and classification of diagnoses.**

encephalitis patients were more likely to by hyponatraemic than other patients (39 [60%] versus 42 [25%], p<0.01; they also had a higher median CSF white cell count (74 [17–200] versus 18 [0–7680] x10$^6$ cells per L, p<0.01). On imaging HSV encephalitis patients were more likely to have abnormal CT scans than those with other forms of encephalitis (24 [41%] of 58 versus 19 [11%] of 166, p<0.001;) and to have abnormal MRI scans (48 [94%] of 51 versus 59 [39%] of 153, p<0.001).

In a multivariate logistic regression analysis having a fever on examination, not having a rash on examination, and lower serum sodium were associated with HSV encephalitis compared to other causes of encephalitis (Table 4). For validity, a sensitivity analysis was performed using imputed data which found minimal difference in the associations found; multivariate analysis demonstrated the same associations. The combined presence of fever, lack of rash, and presence of hyponatremia had a sensitivity of 86% for diagnosis of HSV encephalitis and specificity of 42%, with a positive predictive value (PPV) of 0.36 and negative predictive value of 0.88.

**Table 3. Presenting clinical features and outcomes for 233 patients with encephalitis.**

| | Infective encephalitis 108 | | | | | All Autoimmune 61 | | | | | | |
| | HSV 65 | P value[1] | Other viruses 32 | Other organisms 11 | All infectious causes 108 | Autoimmune Confirmed & probable 38 | p value[3] | Autoimmune possible 23 | All autoimmune 61 | p value[4] | Unknown enceph 87 | All Enceph 233 |
|---|---|---|---|---|---|---|---|---|---|---|---|---|
| Age (med. IQR) | 60 (47,71) | 0.004 | 47 (33, 64) | 62 (47, 62) | 56.5(39, 69) | 38 (24, 59) | 0.004 | 57(38, 64) | 50 (26, 62) | 0.019 | 54 (36, 67) | 54 (34, 68) |
| Female | 38 (59%) | 0.113 | 14 (44%) | 2 (20%) | 54(50%) | 23 (61%) | 0.136 | 13 (57%) | 36 (59%) | 0.108 | 38 (43%) | 115 (49%) |
| White | 64 (99%) | 0.020 | 26 (81%) | 9 (90%) | 99(92%) | 33 (87%) | 1.000 | 21 (91%) | 54(89%) | 0.716 | 78 (90%) | 210 (91%) |
| Co-morbidities (1 or more) | 21 (32%) | 0.757 | 10 (31%) | 5 (50%) | 36(33%) | 7 (18%) | 0.107 | 9 (39%) | 16 (26%) | 0.553 | 36 (41%) | 70 (30%) |
| Immunocompromise | 5 (8%) | 0.299 | 9 (28%) | 2 (20%) | 16(15%) | 1 (3%) | 0.056 | 2 (9%) | 3(5%) | 0.402 | 9 (10%) | 27 (12%) |
| **Symptoms** | | | | | | | | | | | | |
| History of Fever | 46 (71%) | 0.003 | 16 (50%) | 9 (90%) | 71(66%) | 13 (34%) | 0.017 | 9 (39%) | 22(36%) | 0.001 | 43(49%) | 127 (54.5%) |
| Flu like symptoms | 13 (20%) | 1.000 | 4 (13%) | 5 (50%) | 22 (20%) | 5 (13%) | 0.597 | 6 (26%) | 11 (18%) | 0.839 | 14 (16%) | 46 (19.7%) |
| History of Rash | 4 (6%) | 0.166 | 10 (31%) | 0 (0%) | 14 (13%) | 3 (8%) | 0.585 | 3 (13%) | 6 (10%) | 0.791 | 10 (11%) | 27 (11.6%) |
| Severe/worst ever headache | 19 (31%) | 0.635 | 6(19%) | 4 (40%) | 29 (27%) | 3 (8%) | 0.004 | 8 (35%) | 11 (18%) | 0.075 | 31 (36%) | 63 (28.0%) |
| Seizures | 29 (44%) | 0.123 | 6 (19%) | 4 (40%) | 39 (36%) | 21 (55%) | 0.007 | 8 (35%) | 29 (48%) | 0.043 | 24 (28%) | 84 (36.1%) |
| Altered personality/ behaviour | 41 (63%) | 0.370 | 11 (34%) | 0 (0%) | 52 (48%) | 30 (79%) | 0.143 | 17 (17%) | 47 (77%) | 0.119 | 32 (37%) | 159 (68.2%) |
| Agitation | 13 (20%) | <0.001 | 7 (22%) | 5 (50%) | 25 (23%) | 19 (50%) | 0.096 | 13 (57%) | 32 (52%) | 0.012 | 45 (52%) | 89 (38.2%) |
| Lethargy/ increased sleeping | 28 (43%) | 0.111 | 9 (28%) | 1 (10%) | 38 (35%) | 15 (39%) | 0.682 | 7 (30%) | 22 (36%) | 0.862 | 27 (31%) | 80 (34.3%) |
| Psychosis | 1 (2%) | 0.187 | 1 (3%) | 0 (0%) | 2 (2%) | 6 (16%) | 0.006 | 1 (4%) | 7 (12%) | 0.024 | 4 (5%) | 12 (5.2%) |
| Confusion/ disorientation | 52 (80%) | 0.588 | 20 (63%) | 7 (70%) | 79 (73%) | 27 (71%) | 0.543 | 20 (87%) | 47 (77%) | 1.000 | 73 (84%) | 179 (76.8%) |
| Language/ speech problems | 28 (43%) | 0.753 | 13 (41%) | 3 (30%) | 44 (41%) | 17 (45%) | 0.932 | 12 (52%) | 29 (48%) | 0.823 | 45 (52%) | 106 (45.5%) |
| Memory problem | 28 (43%) | 0.527 | 9 (28%) | 2 (20%) | 39 (36%) | 19 (50%) | 0.125 | 9 (39%) | 28 (46%) | 0.262 | 33 (38%) | 91 (39.1%) |
| Hallucinations | 8 (13%) | 1.000 | 3 (9%) | 0 (0%) | 11 (10%) | 6 (16%) | 0.616 | 4 (17%) | 10 (16%) | 0.333 | 12 (14%) | 29 (12.7%) |
| **Examination findings** | | | | | | | | | | | | |
| Fever | 47 (75%) | <0.001 | 11 (34%) | 7 (70%) | 65 (60%) | 7 (18%) | 0.001 | 7 (30%) | 14 (23%) | <0.001 | 30 (34%) | 102 (44.7%) |
| Rash | 4 (6%) | 0.049 | 12 (38%) | 1 (10%) | 17 (16%) | 7 (18%) | 0.623 | 2 (9%) | 9 (15%) | 1.000 | 9 (10%) | 33 (14.2%) |
| GCS ≤12 | 44 (66%) | 0.846 | 18 (56%) | 2 (20%) | 64 (59%) | 11 (29%) | 1.000 | 8 (35%) | 18 (30%) | 1.000 | 20 (23%) | 152 (64%) |
| Abnormal movements | 10 (15%) | 0.160 | 3 (9%) | 1 (10%) | 14 (13%) | 15 (39%) | <0.001 | 9 (39%) | 28 (48%) | <0.001 | 18 (21%) | 52 (22.3%) |
| Focal weakness | 23 (35%) | 0.128 | 8 (25%) | 3 (30%) | 34 (31%) | 5 (13%) | 0.634 | 3 (13%) | 8 (13%) | 0.675 | 18 (21%) | 64 (27.5%) |
| **Investigation findings** | | | | | | | | | | | | |

(*Continued*)

**Table 3.** (Continued)

| | Infective encephalitis 108 | | | | | All Autoimmune 61 | | | | | | |
| | HSV 65 | P value[1] | Other viruses 32 | Other organisms 11 | All infectious causes 108 | Autoimmune Confirmed & probable 38 | p value[3] | Autoimmune possible 23 | All autoimmune 61 | p value[4] | Unknown enceph 87 | All Enceph 233 |
|---|---|---|---|---|---|---|---|---|---|---|---|---|
| Median CSF WCC (IQR) | 74 (17, 200) | <0.01 | 22 (4, 93) | 1480 (117, 3778) | 55 (10, 194) | 8 (1, 32) | <0.01 | 6 (0, 16) | 6 (0, 30) | 0.031 | 21 (4, 116) | 28 (5, 133) |
| Median blood sodium (IQR) | 133 (130, 137) | <0.01 | 138 (133, 140) | 138 (135, 140) | 134 (131, 139) | 139 (138, 142) | 0.910 | 136 (134, 138) | 139 (136, 141) | 0.517 | 138 (134, 141) | 137 (133, 140) |
| Hyponatraemia | 39 (60%) | <0.01 | 11 (34%) | 3 (30%) | 53 (49%) | 5 (13%) | 0.603 | 8 (35%) | 13 (21%) | 0.558 | 23 (26%) | 80/231 (35%) |
| Abnormal initial CT head | 24/58 (41%) | <0.001 | 4/25 (16%) | 2/10 (20%) | 30/103 (29%) | 2 (5%) | 0.041 | 6 (26%) | 8 (13%) | 0.250 | 11/83 (13%) | 43/213 (20%) |
| Abnormal initial MRI head~ | 48/51 (94%) | <0.001 | 10/29 (34%) | 4/8 (50%) | 62/88 (71%) | 14 (37%) | 0.096 | 16 (70%) | 30 (49%) | 1.000 | 31/78 (40%) | 105/200 (53%) |
| Abnormal EEG | 23/29 (79%) | 0.363 | 7/8 (88%) | 4/4 (100%) | 34/41 (83%) | 23/31 (74%) | 0.422 | 13/13 (100%) | 36/44 (82%) | 0.485 | 26/29 (90%) | 83/101 (82%) |
| **Inpatient progress & outcome** | | | | | | | | | | | | |
| ICU admission | 23 (35%) | 1.000 | 7 (22%) | 9 (90%) | 39 (36%) | 20 (53%) | 0.022 | 9 (39%) | 29 (48%) | 0.063 | 46 (53%) | 87 (37%) |
| Hosp LOS Median (IQR) | 34 (21–65) | 0.054 | 16 (10–27) | 39 (15–72) | 26 (16, 55) | 36 (16, 113) | 0.810 | 22 (14, 40) | 26 (15, 80) | 0.987 | 14 (10–25) | 22 (13–49) |
| Mortality admission | 3 (5%) | 0.879 | 1 (3%) | 1 (10%) | 5 (5%) | 3 (8%) | 0.472 | 1 (4%) | 4 (7%) | 0.753 | 4 (5%) | 12 (5%) |
| GOS at discharge | 4 (3, 4.2) | 0.449 | 4 (3, 5) | 5 (3, 5) | 4 (3, 5) | 4 (3, 4.8) | 0.096 | 4 (4, 5) | 4 (3, 5) | 0.324 | 5 (4, 5) | 4 (3,5) |
| **Outcomes at 12 months** | | | | | | | | | | | | |
| Mortality at 12 months | 5 (8%) | 0.576 | 3 (9%) | 1 (10%) | 9 (8%) | 5/30 (17%) | 0.790 | 4/20 (20%) | 9/50 (18%) | 0.674 | 12/84 (14%) | 26 (11%) |
| Median GOS at 12 months | 4 (3, 5) | 0.838 | 5 (4, 5) | 5 (5, 5) | 5 (4, 5) | 5 (3, 5) | 0.515 | 4 (2.5, 5) | 4 (3, 5) | 0.162 | 5 (4, 5) | 5 (3, 5) |

Note: P-value calculated from Chi-square test for categorical variables with >5 individuals in each cell (group combination), Fisher's exact test for categorical variables with <5 individuals in at least one cell (group combination) and one-way ANOVA for continuous variables. P value [1]HSV vs non HSV encephalitis; p value[2] all infective vs not infective encephalitis; p value [3] proven & probable autoimmune vs all other encephalitis; p value[4] all autoimmune vs not autoimmune encephalitis. *>10% missingness of this variable ~In keeping with encephalitis

## Autoimmune encephalitis

The 38 patients with confirmed or probable autoimmune encephalitis typically presented with a few weeks' history (median 30 days [IQR 25–34]) of altered personality and/or confusion, often with agitation and seizures; the most commonly identified antibodies were against

**Table 4. Multivariate logistic regression model with selected variables for HSV encephalitis vs other forms of encephalitis.**

| Selected variables from the model | Parameter estimate | Standard error | Odds ratio | 95% CI of OR | p-value |
|---|---|---|---|---|---|
| Intercept | -0.69 | 0.78 | 0.50 | 0.11, 2.30 | 0.37 |
| Fever on examination | 1.84 | 0.38 | 6.29 | 3.00,13.18 | <0.01 |
| Rash on examination | -1.54 | 0.68 | 0.21 | 0.06,0.81 | 0.02 |
| Presence of Hyponatraemia | 1.11 | 0.37 | 3.04 | 1.48, 6.26 | <0.01 |

Multivariate analysis was performed using 18 clinical and laboratory variables at presentation that were significant from the univariate analysis. The results were fitted by step Bayesian information criterion using backward selection.

**Table 5. Multivariate logistic regression model with selected variables for autoimmune encephalitis (confirmed or probable) versus other forms of encephalitis.**

| Selected variables from the model | Parameter estimate | Standard error | Odds ratio | 95% CI of OR | p-value |
|---|---|---|---|---|---|
| Intercept | -2.06 | 0.78 | 0.13 | 0.03,0.59 | <0.01 |
| Gender (Male) | -1.26 | 0.49 | 0.29 | 0.11,0.75 | 0.01 |
| Abnormal movement | 1.71 | 0.49 | 5.50 | 2.11,14.37 | <0.01 |
| Hyponatraemia | -1.20 | 0.52 | 0.30 | 0.11,0.84 | 0.02 |
| CSF WCC $\geq$20 | -1.31 | 0.45 | 0.27 | 0.11,0.65 | <0.01 |

Multivariate analysis was performed using 19 clinical and laboratory variables at presentation that were significant selected from the univariate analysis. The results were fitted using step Bayesian information criterion using backward selection

VGKC complex and NMDAR (Fig 1). Of note 7 of the 10 VGKC complex positive individuals were tested for LGI1 of which 1 was positive in CSF (serum was not available for retrospective testing). LGI1 testing was not in routine practice at the time of patient recruitment. Patients with confirmed or probable autoimmune encephalitis were younger than the 195 with other forms of encephalitis (median age 38 [IQR 24–59] versus 55 [IQR 37–69] years, p = 0.004) and more likely to be female (23 [61%] versus 90 [46%], p = 0.04). They were more likely to have abnormal movements (15 [39%] versus 27[14%], p<0.01), have seizures (22 [56%] versus 62 [32%], p<0.01), and have a psychosis (6 [15%] versus 6 [3%], p<0.01), but less likely to have a severe headache (3 [8%] versus 60 [32%], p<0.01) than those with other forms of encephalitis. They were also less likely to be hyponatraemic, and had a lower median CSF white cell count (8 [IQR 1–32] versus 42 [8–168] x10$^6$ cells per L, p<0.01) than those with other causes (Table 3).

In a multiple logistic regression, female sex, abnormal movements, lack of hyponatremia and a CSF white cell count <20 x10$^6$/L were associated with confirmed or probable autoimmune encephalitis, compared to other causes (Table 5). The presence of all these features gave a sensitivity of 97%, specificity 12%, with positive predictive value (PPV) of 0.18 and negative predictive value (NPV) of 0.96.

Because new diagnostic guidelines have been introduced which include case definitions for possible, as well as probable and confirmed autoimmune encephalitis, we repeated the analysis including 23 such patients. This showed that the 61 patients with confirmed, probable or possible autoimmune encephalitis were likely to be younger and have seizures, psychosis, abnormal movements, and a lower CSF white cell count, than those with other forms of encephalitis (Table 3); in addition, they were more likely to have agitation.

In a multivariate logistic regression analysis, a lack of fever on examination, the presence of abnormal movements, and a CSF white cell count of <20 cells x10$^6$/L were associated with a diagnosis of confirmed, probable or possible autoimmune encephalitis, compared to other causes (S1 Table). For validity, a sensitivity analysis was performed using imputed data which found minimal difference in the associations found; multivariate analysis demonstrated the same associations. The combined absence of fever, presence of abnormal movements and CSF white cell count <20 cells x10$^6$/L had a sensitivity of 95% for diagnosis of confirmed, probable or possible autoimmune encephalitis and specificity of 37%, with a PPV of 0.35 and NPV of 0.95.

## Treatment and outcomes

276 patients with suspected encephalitis had aciclovir initiated a median 15 hours (IQR 5–44) after hospital admission; this included the 61 patients with HSV encephalitis whose time to treatment was 14 hours (IQR 5–50). Time to treatment with aciclovir improved during the

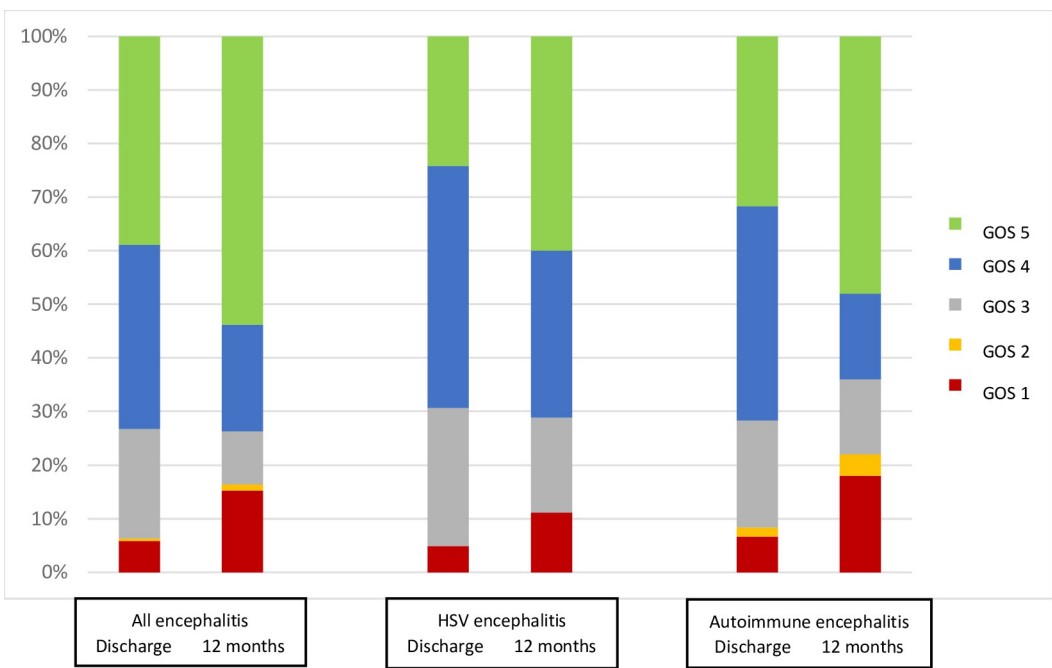

**Fig 2. Glasgow outcome scale scores at discharge and at 12 months after discharge for all 233 patients with encephalitis, 64 with HSV encephalitis and 38 with autoimmune encephalitis.**

course of the study from 23 hours (IQR 10–56) in the first year, to 12 hours (IQR 4–28) in the 2nd and 10 hours (IQR 3–34) in the 3rd year. Among the 38 patients treated for autoimmune encephalitis the median time to first immunosuppressant drug (corticosteroids in 28, intravenous immunoglobulin in 6) was 125 hours (IQR, 45–250). This did not change significantly across the course of the study.

Overall, 87 (37%) of the 233 patients were admitted to intensive care; those with confirmed or probable autoimmune encephalitis were more likely to be admitted than those with other diagnoses (20 [53%] of 38 verses 67 [34%] of 195, p = 0.022). 12 (5%) of 233 patients died in hospital and 86 (37%) made a full recovery; the median Glasgow Outcome Scale score at discharge was 4; these outcomes did not differ significantly between groups (Fig 2). By 12 months the mortality had increased to 26 (11%) and the number with full recovery to 92 (54%) of 171 with the information available, so that the median Glasgow Outcome Scale score had increased to 5.

## Discussion

This is one of the largest prospective studies on unselected encephalitis patients from any western industrialised nation, recruiting more patients with HSV encephalitis than any previous prospective report. We found just under half the patients had an infectious cause, approximately one quarter had a confirmed, probable or possible autoimmune encephalitis, and one quarter had no aetiology identified. Over the last 20 years the epidemiology of encephalitis has changed considerably. Whilst HSV continues to be the most important sporadic cause, patients with autoimmune aetiologies are being recognised increasingly. Encephalitis remains relatively rare overall, with an incidence of 2–5 per 100,000 per year; [24] however, it causes a disproportionately high disease burden because of the devastating impact of this acquired brain injury on those affected, particularly if there are delays in recognition, diagnosis and treatment.

For HSV encephalitis, the link between worse outcome and delays in treatment, particularly beyond 48 hours after hospital admission, has been recognised for some years [10, 25]. National guidelines for viral encephalitis were developed in the UK in 2012 [17]. Their introduction through an NIHR programme grant in applied research was associated with a reduction in the time to aciclovir treatment for those with HSV encephalitis. The programme examined where delays occur, introduced innovative measures such as a lumbar puncture pack to streamline management [15, 26], and included intense efforts to raise awareness of encephalitis and the national guidelines, in conjunction with The Encephalitis Society, our partner in patient and public involvement and engagement [27].

Across the three years of our study, we found the median time from admission to treatment reduced from 23 hours to ten hours, which is still longer than the 6 hours recommended by the national guidelines. Our findings that fever on examination, absence of a rash, and hyponatremia were associated with a diagnosis of HSV encephalitis may help improve things. A small retrospective study from Argentina suggested fever, headache, and higher CSF white cell count and protein were more likely in infectious than non-infectious encephalitis [28]; in a study from Turkey hyponatremia was associated with HSV encephalitis [29].

A retrospective comparison of 95 patients from China found involuntary movements and memory deficits were more common in autoimmune than infectious encephalitis patients [30]. A similar retrospective study in Australia evaluated retrospectively 84 patients evaluated over a ten-year period and found headache, fevers, altered consciousness, psychiatric symptoms, and a CSF pleocytosis were more common in infectious encephalitis, whereas seizures were more common in autoimmune [31].

In practice, the typical management for patients with suspected encephalitis is to perform a lumbar puncture and neuroimaging whist starting intravenous aciclovir and antibiotics. If an infectious cause is confirmed by PCR or culture of the cerebrospinal fluid, the antimicrobials are adjusted accordingly. The dilemma clinicians face currently is what to do when the investigations for infectious causes are negative. The results of tests for antibodies associated with autoimmune encephalitis, such as NMDAR, LGI-1, and GAD, often take several days to come back, and may not be helpful in seronegative autoimmune cases, which are being recognised increasingly [22, 32]. In our study the median time from hospital admission to immunosuppressive therapy was 5 days.

Our study found female sex, abnormal movements, a lack of hyponatremia and a CSF white cell count <20 x10$^6$/L were associated with confirmed or probable autoimmune encephalitis. These features have also been reported in studies of individual causes of autoimmune encephalitis: female sex, abnormal movements and a moderate pleocytosis are reported for NMDAR [33], a moderate pleocytosis for LGI-1 [34], and female sex for GAD-associated encephalitis [9]. Interestingly, other features reported frequently for individual forms of autoimmune encephalitis, such as abnormal mental status and seizures, were not helpful in distinguishing autoimmune from other causes of encephalitis in our study. Although neurologists focus on the clinical features which can distinguish between different causes of autoimmune encephalitis, for the general and infectious disease clinician, the critical thing is to identify patients who may have any form of autoimmune encephalitis, and thus benefit from prompt initiation of presumptive treatment with immunosuppressive drugs. We thus analysed all patients with probable or confirmed autoimmune encephalitis as a single group, whatever the underlying aetiology. There are very few such prospective studies of non-selected patients with encephalitis of all causes. A retrospective comparison of 95 patients from China found involuntary movements and memory deficits were more common in autoimmune than infectious encephalitis patients [30]. A similar retrospective study evaluated 84 encephalitis patients in Austria admitted over a ten-year period, including 34 with confirmed infection and 17 with

confirmed autoimmune disease; headache, fevers, altered consciousness, psychiatric symptoms, and a CSF pleocytosis were more common in infectious encephalitis, whereas seizures were more common in autoimmune patients [31].

An expert review concluded that we are too reliant on antibody testing, and need syndrome-based diagnostics of possible, probable and confirmed encephalitis to allow prompt initiation of therapy [22]. Our study preceded these guidelines, but we modified our classification to include patients with possible autoimmune encephalitis, and repeated our analysis including such patients. This confirmed the importance of abnormal movements and a moderate pleocytosis in identifying patients with autoimmune disease. Although there are data to suggest earlier treatment is beneficial in patients with proven autoimmune encephalitis, at what stage to start treatment in patients with suspected autoimmune encephalitis is less clear, and there are no randomised trials [22].

The diagnosis of antibody-associated encephalitis for most of our patients was based on antibody detection in the serum, rather than CSF. For GAD encephalitis in particular this may be important, because GAD antibodies occur at 1% of healthy people and in 80% of those with type 1 diabetes; there is not yet convincing evidence that GAD antibodies are pathogenic, rather than an epiphenomenon [9]. However, none of GAD positive patients in our study had diabetes, and all were felt to need immunosuppressive drugs, as judged by neurologists.

The presence of female sex, abnormal movements, lack of hyponatremia plus a CSF white cell count <20 x10$^6$/L had a sensitivity of 95% for diagnosis of autoimmune encephalitis and specificity of 37%, with a PPV of 0.35 and NPV of 0.95. The high sensitivity and high negative predictive value are what is required to minimise the chance of overlooking patients with the condition. Such indicators should raise a strong suspicion, and might be used to guide clinicians to start presumptive corticosteroid treatment for patients, as encouraged by the recent guidelines [22]. A prospective study is needed to evaluate them fully.

With increasing recognition of autoimmune encephalitis, and the wider availability of antibody testing, the time to initiation of treatment is likely to have improved since our study was completed. Corticosteroids are the first line therapy for autoimmune encephalitis, and additional immunosuppressive treatment is recommended for some forms, but there have been few randomised trials. A prospective placebo-controlled trial of intravenous immunoglobulin in autoimmune encephalitis is currently recruiting in the UK.

Our study had several limitations. Firstly, although ours was one of the largest prospective studies of unselected patients with encephalitis, the size was limited, especially in terms of numbers with specific autoimmune diagnoses and also inherently biased towards those autoimmune causes that would present acutely or subacutely whereas some forms are known to take more insidious and longer presentation courses. Secondly, because it was performed some years ago, the number of patients with some of the rarer antibodies, was limited. We re-evaluated samples as newer antibody tests became available, but applying tests retrospectively to samples which may have been frozen and thawed many times is never ideal. There may also have been a bias towards identifying well recognised forms of encephalitis, such as HSV, rather than newer emerging forms of autoimmune disease. The time to initiation of treatment for autoimmune encephalitis may have reduced since our study, due to increasing recognition of the condition. We were also reliant on testing of serum for the most part rather than CSF. We now know that antibody testing of CSF has greater specificity. Although our primary purpose was identifying patients with any form of autoimmune encephalitis, there is heterogeneity among different types of autoimmune encephalitis, and grouping them together will have blunted these differences.

In summary, we have shown that among adults admitted to hospital with a diagnosis of encephalitis, clinical features can be used to identify those with HSV and those with

autoimmune disease to guide presumptive treatment. Currently, the initiation of treatment for patients with HSV is significantly quicker than for those with autoimmune disease. The presence of female sex, abnormal movements, lack of hyponatremia and a CSF white cell count $<20$ x10$^6$/L predicted autoimmune encephalitis with high sensitivity and high negative predictive value and may be used to guide presumptive treatment, but this requires further evaluation in a prospective study.

## Supporting information

**S1 Table. Multivariate logistic regression model with selected variables for autoimmune encephalitis (confirmed, probable and possible) versus other forms of encephalitis.**
(DOCX)

**S2 Table. Presenting clinical features and outcomes of the autoantibody mediated forms of autoimmune encephalitis.**
(DOCX)

**S1 Data.**
(XLSX)

## Acknowledgments

The authors would like to thank all the participants who gave their time to help with this study, and the research nurses and principal investigators at the hospital sites for their help with recruitment. We would also like to thank Chloe Smith, Greg Gibson, Rebecca Spencer, Mike Scully, Kieran Crabtree, Katy Smith and Chris Jecks for ensuring data collection was entered from the sites.

ENCEPH-UK Study group lead author Tom Solomon[1]

Contact email for group lead: tsolomon@liverpool.ac.uk.

ENCEPH-UK study group members; Ruth Backman[1], Gus Baker[2], Nicholas J Beeching[3,4], Rachel Breen[5], David Brown[6], Chris Cheyne[7], Enitan D Carrol[1,8], Nicholas W S Davies[9], Sylviane Defres[1,3,4], Ava Easton[10], Martin Eccles[11], Robbie Foy[12], Marta Garcia-Finana[7], Julia Granerod[6], Julia Griem[13], Michael Griffiths[1,8], Alison Gummery[1], Lara Harris[13], Helen Hickey[5], Helen Hill[5], Ann Jacoby[2], Hayley Hardwick[1], Ciara Kierans[14], Michael Kopelman[13], Rachel Kneen[1,8], Gill Lancaster[15], Michael Levin[16], Rebecca McDonald[17], Antonieta Medina-Lara[18], Esse Menson[19], Benedict Michael[1], Natalie Martin[20], Andrew Pennington[17], Andrew Pollard[20], Julie Riley[17], Manish Sadarangani[20] Anne Salter[21], Kukatharmini Tharmaratnam[7] Maria Thornton[17], Angela Vincent[22], Charles Warlow[23].

1. Department of Clinical Infection Microbiology and Immunology, Institute of Infection and Global Health, University of Liverpool, Liverpool, UK

2. Department of Clinical Neuropsychology, The Walton Centre NHS Foundation Trust, Liverpool, UK

3. Tropical and Infectious Disease Unit, Liverpool University Hospitals NHS Foundation Trust, Liverpool, UK

4. Clinical Trials Unit, Liverpool, UK

5. Department of Clinical Sciences, Liverpool School of Tropical Medicine, Liverpool, UK

6. Public Health England (formerly Health Protection Agency) Colindale, London, UK

7. The Department of Biostatistics, Institute of translational medicine, University of Liverpool, Liverpool, UK

8. Alder Hey Hospital Children's NHS Foundation Trust, Liverpool, UK

9. Department of Neurology, Chelsea and Westminster NHS Trust, London, UK

10. Encephalitis Society, Malton, UK

11. Faculty of Medical Sciences, Institute of Health and Society, Newcastle University, Newcastle, UK

12. Faculty of Medicine and Health, Institute of Health Sciences, Leeds University, Leeds, UK

13. Institute of Psychiatry, Kings College, London, UK

14. Public Health and Policy, Institute of psychology Health and Society, University of Liverpool, Liverpool, UK

15. Mathematics and Statistics, Lancaster University, Lancaster, UK

16. Paediatrics and International Child Health, Imperial College, London, UK

17. Research and Development Department, The Walton Centre NHS Foundation Trust, Liverpool, UK

18. Health Economics Group, University of Exeter medical School, Exeter, UK

19. Infectious diseases and Immunology team, Evelina London Children's Hospital, London, UK

20. Oxford Vaccine Group, University of Oxford, Oxford, UK

21. Patient representative, Encephalitis Society, Malton, UK

22. Nuffield Department of Clinical Neurosciences, University of Oxford, Oxford, UK

23. Department of Neurosciences, Western General Hospital, University of Edinburgh, Edinburgh, UK

Hospitals sites involved in recruitment into ENCEPH UK

Principal Investigators

Gavin Barlow[1], Nicholas J Beeching[2], Thomas Blanchard[3], Richard Body[4], Gavin Boyd[5], Lucia Cebria-Prejan[6], David Chadwick[7], Richard Cooke[8] Pamela Crawford[9], Brendan Davies[10], Nicholas W S Davies[11], Sam Douthwaite[12], Hedley Emsley[13], Simon Goldenberg[12], Clive Graham[14], Steve Green[15], Clive Hawkins[10], Dianne Irish[16], Kate Jeffrey[17], Matt Jones[18], Liza Keating[19], Jeff Keep[20], Susan Larkin[8], Maria Leita[17], Derek Macallan[21], Jane Minton[22], Kavya Mohandas[23], Ed Moran[24], David Muir[25], Monicka Pasztor[26], Matthew Reed[27], Tom Solomon[28], Philip Stanley[29], Julian Sutton[30], Peter Thomas[31], Guy Thwaites[12], John Weir[19], Mark Zuckerman[20].

1. Hull and East Yorkshire Hospitals NHS Trust, Hull.

2. Liverpool University Hospitals NHS Foundation Trust Liverpool

3. North Manchester General Hospital, Manchester

4. Manchester Royal Infirmary, Manchester

5. Calderdale and Huddersfield NHS Foundation Trust, Huddersfield

6. Mid Yorkshire Hospitals NHS Trust, Pinderfields

7. South Tees Hospitals NHS Foundation Trust, Middlesbrough

8. Aintree University Hospital NHS Foundation Trust, Liverpool

9. York Teaching Hospital NHS Foundation Trust, York

10. University Hospitals North Midlands, Stoke on Trent

11. Chelsea and Westminster Hospital NHS Foundation Trust, London

12. Guy's and St Thomas' NHS Foundation Trust, London

13. Lancashire Teaching Hospitals NHS Foundation Trust, Royal Preston Hospital, Preston

14. North Cumbria University Hospitals NHS Trust, Carlisle

15. Sheffield Teaching Hospitals NHS Foundation Trust, Sheffield

16. Royal Free London NHS Foundation Trust, London

17. Oxford University Hospitals NHS Foundation Trust, John Radcliffe Hospital, Oxford

18. Salford Royal NHS Foundation trust, Salford

19. Royal Berkshire NHS Foundation Trust, Reading

20. Kings College Hospital NHS Foundation Trust, London

21. St George's University Hospitals NHS Foundation Trust, London

22. Leeds Teaching Hospitals NHS Trust, St James' University Hospital, Leeds

23. Wirral University Teaching Hospital NHS Foundation Trust–Arrowe Park Hospital, Upton

24. University Hospitals Birmingham NHS Foundation Trust, Birmingham Heartlands Hospital, Birmingham

25. Imperial College Healthcare NHS Trust, St Mary's Hospital, London

26. University Hospitals of Morecambe Bay NHS Foundation Trust, Lancaster

27. NHS Lothian, Royal Infirmary Edinburgh, Edinburgh

28. The Walton Centre NHS Foundation Trust, Liverpool

29. Bradford Teaching Hospitals NHS Foundation Trust, Bradford

30. University Hospital Southampton NHS Foundation Trust, Southampton

31. Milton Keynes University Hospital NHS Foundation Trust, Milton Keynes

## Author Contributions

**Conceptualization:** Sylviane Defres, Benedict D. Michael, Nicholas W. S. Davies, Ava Easton, Michael J. Griffiths, Rachel Kneen, Antonieta Medina-Lara, Anne Christine Salter, Nicholas J. Beeching, Enitan Carrol, Angela Vincent, Marta Garcia-Finana, Tom Solomon.

**Data curation:** Sylviane Defres, Maneesh Bhojak, Kumar Das, Hayley Hardwick, Chris Cheyne, Antonieta Medina-Lara, Nicholas J. Beeching, Enitan Carrol, Angela Vincent, Marta Garcia-Finana, Tom Solomon.

**Formal analysis:** Sylviane Defres, Kukatharmini Tharmaratnam, Michael J. Griffiths, Maneesh Bhojak, Kumar Das, Antonieta Medina-Lara, Angela Vincent, Marta Garcia-Finana, Tom Solomon.

**Funding acquisition:** Nicholas W. S. Davies, Ava Easton, Michael J. Griffiths, Rachel Kneen, Antonieta Medina-Lara, Nicholas J. Beeching, Enitan Carrol, Angela Vincent, Marta Garcia-Finana, Tom Solomon.

**Investigation:** Sylviane Defres, Mark Ellul, Nicholas J. Beeching, Enitan Carrol, Angela Vincent, Marta Garcia-Finana, Tom Solomon.

**Methodology:** Sylviane Defres, Benedict D. Michael, Mark Ellul, Nicholas W. S. Davies, Ava Easton, Michael J. Griffiths, Maneesh Bhojak, Kumar Das, Chris Cheyne, Rachel Kneen, Antonieta Medina-Lara, Anne Christine Salter, Nicholas J. Beeching, Enitan Carrol, Angela Vincent, Marta Garcia-Finana, Tom Solomon.

**Project administration:** Sylviane Defres, Ava Easton, Hayley Hardwick, Nicholas J. Beeching, Enitan Carrol, Tom Solomon.

**Resources:** Ava Easton, Hayley Hardwick, Antonieta Medina-Lara, Enitan Carrol, Tom Solomon.

**Supervision:** Kumar Das, Angela Vincent, Marta Garcia-Finana, Tom Solomon.

**Validation:** Sylviane Defres, Kukatharmini Tharmaratnam, Nicholas W. S. Davies.

**Writing – original draft:** Sylviane Defres.

**Writing – review & editing:** Sylviane Defres, Kukatharmini Tharmaratnam, Benedict D. Michael, Mark Ellul, Nicholas W. S. Davies, Ava Easton, Michael J. Griffiths, Maneesh Bhojak, Kumar Das, Hayley Hardwick, Chris Cheyne, Rachel Kneen, Antonieta Medina-Lara,

Anne Christine Salter, Nicholas J. Beeching, Enitan Carrol, Angela Vincent, Marta Garcia-Finana, Tom Solomon.

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
