## [Decision Letter · Decision Letter 0]

14 Jun 2022

PONE-D-21-37215Clinical predictors of encephalitis in UK adults – a multi-centre prospective observational cohort studyPLOS ONE

Dear Dr. Defres,

Thank you for submitting your manuscript to PLOS ONE. Firstly, we would like to apologize for the delay in processing your manuscript. It has been exceptionally difficult to secure reviewers to evaluate your study. We have now received one completed review, which is available below.

Please note that we have only been able to secure a single reviewer to assess your manuscript. We are issuing a decision on your manuscript at this point to prevent further delays in the evaluation of your manuscript. Please be aware that the editor who handles your revised manuscript might find it necessary to invite additional reviewers to assess this work once the revised manuscript is submitted. However, we will aim to proceed on the basis of this single review if possible.

After careful consideration, we feel that it has merit but does not fully meet PLOS ONE’s publication criteria as it currently stands. Therefore, we invite you to submit a revised version of the manuscript that addresses the points raised during the review process.

We look forward to receiving your revised manuscript.

Kind regards,

Thomas Tischer

Staff Editor

PLOS ONE

Journal Requirements:

3. Please amend the manuscript submission data (via Edit Submission) to include author Michael J Griffiths.

4. One of the noted authors is a group or consortium [insert name of group or team]. In addition to naming the author group, please list the individual authors and affiliations within this group in the acknowledgments section of your manuscript. Please also indicate clearly a lead author for this group along with a contact email address.

Reviewers' comments:

Reviewer's Responses to Questions

**Comments to the Author**

1. Is the manuscript technically sound, and do the data support the conclusions?

Reviewer #1: Yes

2. Has the statistical analysis been performed appropriately and rigorously? 

Reviewer #1: Yes

3. Have the authors made all data underlying the findings in their manuscript fully available?

Reviewer #1: Yes

4. Is the manuscript presented in an intelligible fashion and written in standard English?

Reviewer #1: Yes

5. Review Comments to the Author

Reviewer #1: The authors describe a large prospective, multicentric cohort of patients with suspected encephalitis between 2012 and 2015. They screened 794 patients, had to exclude 453 and recruited 341 patients with suspected encephalitis. 233 had encephalitis (according to prespecified criteria) of whom 97 were viral, 11 bacterial/fungal, 61 autoimmune and 64 unknown. Their main findings were that patients with HSV more often had fever and were older (and has less frequently rash) while patients categorized as definite or probable AE were more often female, had movement disorders and had less often fever and less pleocytosis in cSF.

In general the authors have recruited a very large cohort including infectious and autoimmune encephalitis allowing for comparative studies. The manuscript is well written and especially the discussion is nice to read. It openly discusses all obvious limitations. The main findings are well supported by their data and the figures and tables are instructive. It is a bit surprising, that the results get published only now.

My main issues are:

- Antibody studies. Indeed the authors acknowledge that their antibody studies evolved during the trial with new antibodies described… However, it appears the authors had stored serum available and it would have been possible to comprehensively reexamine the patients for common and less common antibodies. Freezing/thawing cycles usually are not a problem for serologic studies. I think it is somewhat of a missed chance to identify AE antibody frequency in this unbiased, remarkable cohort of acute encephalitis. The lack of CSF is unfortunate, the advantage of CSF testing is correctly pointed out in the discussion. On the same note, the cited Graus et al criteria emphasize that the category of probable seronegative AE should only be entertained after comprehensive testing, which the authors did not do. Finally, LGI1/VGKC is not a valid category anymore. How was the antibody tested? VGKC plus CBA or only VGKC. VGKC testing alone will be difficult to classify here.

- It would be instructive (as supplementary data) to know the features of the ab defined subgroups of AE even though numbers are small. This also holds true for outcome. E.g. In suspect that the majority of movement disorders stems from the NMDARE subgroup. Gad and Hashimoto rarely have MD.

- I would suggest to point out in the discussion, that the spectrum of AE is biased by the intial criteria chosen. Patients with rapid/subacute onset. We know that Iglon5, CAPSR2, LGI1 can have very extended and sometimes insidious onset. Therefore, the spectrum of AE is more representative of acute rapid onset AE and not AE in general.

- The findings that HSVI is associated with lack of rash. Is that mainly carried by the 9 VZV encephalitis patients? The other viruses usually do not associate with rash as far as I know.

6. PLOS authors have the option to publish the peer review history of their article (what does this mean?). If published, this will include your full peer review and any attached files.

Reviewer #1: No

---

## [Author Response · Author response to Decision Letter 0]

5 Sep 2022

all points covered in response to reviewer letter

---

## [Decision Letter · Decision Letter 1]

12 Oct 2022

PONE-D-21-37215R1Clinical predictors of encephalitis in UK adults – a multi-centre prospective observational cohort studyPLOS ONE

Dear Dr. Defres,

Thank you for submitting your manuscript to PLOS ONE. After careful consideration, we feel that it has merit but does not fully meet PLOS ONE’s publication criteria as it currently stands. Therefore, we invite you to submit a revised version of the manuscript that addresses the points raised during the review process.

I would appreciate you addressing the two remaining issues raised by the reviewer - if addressed satisfactorily, I would hope we could move to acceptance.==============================

We look forward to receiving your revised manuscript.

Kind regards,

Richard John Lessells, BSc, MBChB, MRCP, DTM&H, DipHIVMed, PhD

Academic Editor

PLOS ONE

Journal Requirements:

Reviewers' comments:

Reviewer's Responses to Questions

**Comments to the Author**

1. If the authors have adequately addressed your comments raised in a previous round of review and you feel that this manuscript is now acceptable for publication, you may indicate that here to bypass the “Comments to the Author” section, enter your conflict of interest statement in the “Confidential to Editor” section, and submit your "Accept" recommendation.

Reviewer #1: (No Response)

2. Is the manuscript technically sound, and do the data support the conclusions?

Reviewer #1: Partly

3. Has the statistical analysis been performed appropriately and rigorously? 

Reviewer #1: Yes

4. Have the authors made all data underlying the findings in their manuscript fully available?

Reviewer #1: Yes

5. Is the manuscript presented in an intelligible fashion and written in standard English?

Reviewer #1: Yes

6. Review Comments to the Author

Reviewer #1: The authors have addressed most of the issues I raised. Unfortunately, I am not fully satisfied:

1.The authors comment to the reviewer, that indeed of 10 VGKC pos patients, only 5 were tested for LGI1 and none were positive. Therefore it is clearly misleading to label this group as VGKC/LGI1 patients, which implies positivity against LGI1. Referring to PMID: 32595134 from Sophia Michaels, Paddy Waters and Sarosh Irani, LGI1/CASPR2 negative VGKC findings are “more often misleading than not.”. Yet, the authors still introduce the term VGKC complex in the introduction and include this group which is – frankly – not appropriate anymore in 2022 even though the data and testing was done several years ago. And the lack of LGI1 positivity in the group tested for it (n=5) is not clearly stated in the manuscript. Unless there is serum for retesting available, I would suggest to include that entire group (n=10) into the possible AE (if they fulfill the critertia) but not in the definite group according to the Graus et al consensus paper.

2.The authors have clarified their testing strategy. Indeed all patients had NMDAR and GAD (and VGKC) testing and eventually patients were also tested for a panel of newer cell surface antibodies. To make this clear for readers, I would add the number of patients CSF and serum only tested with core testing and the ones tested with the extended panel (either during clinical routine or during re-testing of stored samples). This would make it easier to judge the fraction of idiopathic and possible cases that might contain undetected abs like cASPR2. Also please indicate the panel of “paraneoplastic” onconeural abs that were tested.

Minor comment: In figure 1 Panel Autoimmune: Iglon5 instead of IglonS, paraneoplastic (seronegative) cases

7. PLOS authors have the option to publish the peer review history of their article (what does this mean?). If published, this will include your full peer review and any attached files.

Reviewer #1: No

---

## [Author Response · Author response to Decision Letter 1]

11 Jan 2023

these are covered in the response to reviewers cover letter

---

## [Decision Letter · Decision Letter 2]

20 Feb 2023

Clinical predictors of encephalitis in UK adults – a multi-centre prospective observational cohort study

PONE-D-21-37215R2

Dear Dr. Defres

We’re pleased to inform you that your manuscript has been judged scientifically suitable for publication and will be formally accepted for publication once it meets all outstanding technical requirements.

Kind regards,

Rizaldy Taslim Pinzon

Academic Editor

PLOS ONE

Additional Editor Comments (optional):

This is a good study. There is novelty in this study. Thank you for your prompt reply and responses to the reviewer.

Reviewers' comments:

Reviewer's Responses to Questions

**Comments to the Author**

1. If the authors have adequately addressed your comments raised in a previous round of review and you feel that this manuscript is now acceptable for publication, you may indicate that here to bypass the “Comments to the Author” section, enter your conflict of interest statement in the “Confidential to Editor” section, and submit your "Accept" recommendation.

Reviewer #1: All comments have been addressed

2. Is the manuscript technically sound, and do the data support the conclusions?

Reviewer #1: Yes

3. Has the statistical analysis been performed appropriately and rigorously? 

Reviewer #1: Yes

4. Have the authors made all data underlying the findings in their manuscript fully available?

Reviewer #1: Yes

5. Is the manuscript presented in an intelligible fashion and written in standard English?

Reviewer #1: Yes

6. Review Comments to the Author

Reviewer #1: The reviewers have addressed all comments. I would recommend accepting the article. It is a relevant contribution to the field.

7. PLOS authors have the option to publish the peer review history of their article (what does this mean?). If published, this will include your full peer review and any attached files.

Reviewer #1: No

---

## [Editor Report · Acceptance letter]

10 May 2023

PONE-D-21-37215R2 

Clinical predictors of encephalitis in UK adults – a multi-centre prospective observational cohort study 

Dear Dr. Defres:

I'm pleased to inform you that your manuscript has been deemed suitable for publication in PLOS ONE. Congratulations! Your manuscript is now with our production department. 

Kind regards, 

on behalf of

Dr. Rizaldy Taslim Pinzon 

Academic Editor

PLOS ONE